# Scratch-induced partial skin wounds re-epithelialize by sheets of independently migrating keratinocytes

Laura Bornes[1], Reinhard Windoffer[2], Rudolf E Leube[2], Jessica Morgner[1,*], Jacco van Rheenen[1,*]

**Re-epithelialization is a crucial process to reestablish the protective barrier upon wounding of the skin. Although this process is well described for wounds where the complete epidermis and dermis is damaged, little is known about the re-epithelialization strategy in more frequently occurring smaller scratch wounds in which structures such as the hair follicles and sweat glands stay intact. To study this, we established a scratch wound model to follow individual keratinocytes in all epidermal layers in the back skin of mice by intravital microscopy. We discover that keratinocytes adopt a re-epithelialization strategy that enables them to bypass immobile obstacles such as hair follicles. Wound-induced cell loss is replenished by proliferation in a distinct zone away from the wound and this proliferation does not affect overall migration pattern. Whereas suprabasal keratinocytes are rather passive, basal keratinocytes move as a sheet of independently migrating cells into the wound, thereby constantly changing their direct neighboring cells enabling them to bypass intact obstacles. This re-epithelialization strategy results in a fast re-establishment of the protective skin barrier upon wounding.**

## Introduction

Re-epithelialization of the skin is a crucial process during healing of epidermal wounds to restore the barrier function of the skin. This repair is well studied in the so-called full-thickness wounds, for example, upon punch biopsies, in which epidermis and dermis of the skin are lost. These wounds heal in three phases. The inflammatory phase is initiated immediately after wounding. It is characterized by the activation and infiltration of inflammatory cells and the formation of a blood clot. This phase is followed by the regeneration phase where the epidermis re-epithelializes, and depending on the wound size, forms a granulation tissue consisting of connective tissue and sprouting microvasculature. During the last phase, epidermis, dermis, blood vessels, and the extracellular matrix remodel to restore skin homeostasis (Gurtner et al, 2008;

Pastar et al, 2014; Dekoninck & Blanpain, 2019). Although these phases are well characterized, they only describe repair of deep wounds. Most daily life skin wounds, however, are caused by scratching, scraping, or superficial cuts. These wounds damage the epidermis and parts of the underlying dermis leaving hair follicles and sweat glands intact. In contrast to full-thickness wounds, partial-thickness wounds can heal by re-epithelialization without the requirement of granulation tissue formation. The healing process is therefore much faster in partial-thickness wounds than in full-thickness wounds (Pastar et al, 2014; Rittie, 2016) and most likely requires a different re-epithelialization strategy which is largely unknown. In full-thickness wounds, re-epithelialization is a multi-day process, depending on the wound size, during which keratinocytes not only migrate to cover the wound bed but, instead, the entire epidermis coordinates wound closure and re-establishment of epidermal homeostasis by migration, proliferation, and differentiation simultaneously (Aragona et al, 2017; Park et al, 2017). The mode of re-epithelialization and keratinocyte migration has mostly been studied in ex vivo and in vitro models, as well as in static analyses after full-thickness wounding in animal models (Laplante et al, 2001; Sorg et al, 2007; Safferling et al, 2013; Wang et al, 2013). Ex vivo and in vitro models do not comprise the entire complexity of cellular cross talks between different cell types in the skin and neglect systemic factors that affect wound healing in vivo (Werner & Grose, 2003). Three different models of how keratinocytes migrate and re-epithelialize a wound have been proposed. The leap-frog model ("leap-frogging" [Krawczyk, 1971; Lambert et al, 1984; Usui et al, 2005]) suggests that suprabasal keratinocytes are the main cell type that close the wound by sliding over leading basal keratinocytes, rolling into the wound bed and forming contact with the basement membrane to cover up the wound site. This model is opposed by the model of collectively migrating multilayers. According to this model, both basal and suprabasal keratinocytes simultaneously migrate unidirectionally and tightly connected across the wound edge until an intact epidermal sheet is re-established (Farooqui & Fenteany, 2005; Poujade et al, 2007). By contrast, the sliding model ("tractor-tread" [Woodley, 1988; Poujade et al, 2007]) postulates that only basal keratinocytes at the leading edge migrate into the wound bed and drag the physically connected

[1]Division of Molecular Pathology, Oncode Institute, Netherlands Cancer Institute, Amsterdam, The Netherlands   [2]Institute of Molecular and Cellular Anatomy, Rheinisch-Westfälische Technische Hochschule Aachen University, Aachen, Germany

Correspondence: j.morgner@nki.nl; j.v.rheenen@nki.nl
*Jessica Morgner and Jacco van Rheenen contributed equally to this work

suprabasal sheet with them until wound closure is established (Friedl & Gilmour, 2009; Matsubayashi et al, 2011). Although there is support for all three predicted models, the mode of keratinocyte dynamics and their behavior in vivo upon partial-thickness wounding without the formation of granulation tissue has not been elucidated. Here, we establish an in vivo epidermal scratch wounding model in the back skin of a fluorescent mouse model that enables us to study the dynamics of keratinocyte behavior during re-epithelialization during the entire wound healing process by intravital microscopy. We show that only basal keratinocytes migrate towards the wound, whereas suprabasal cells remain in place. Basal keratinocytes move independently within a collectively migrating sheet and surpass hair follicles by disassembly and re-establishment of cell–cell contacts with surrounding miscellaneous basal keratinocytes. The loss of cells due to damage is replenished by proliferation in a distinct zone away from the wound site and proliferation does not affect overall migration pattern.

## Results and Discussion

### Establishment of a murine scratch wounding model to investigate epidermal dynamics during wound healing in vivo

To investigate the dynamics of both individual cells and population of cells during epidermal re-epithelialization of superficial wounds in vivo, we imaged wound healing in R26-mTmG; R26-ACTB-CreERT2 mice. These mice express membrane-targeted Tomato (mTom[+]) in all cells facilitating the visualization of the entire cell population in the skin. Upon administration of low doses of tamoxifen, dispersed single cells change their color and start expressing membrane-targeted GFP (GFP[+]) enabling us to study the behavior of individual cells (Fig 1A). We induced scratch wounds in the back skin of these mice and monitored mTom[+] and GFP[+] cells by intravital microscopy from the moment of wounding until wound closure which took ~1–2 d (Video 1). With this approach, all epidermal mTom[+] cells (colored in magenta) as well as individual GFP[+] cells (colored in green) can be identified and tracked over time. The scheme in Fig 1B illustrates main features of this approach (Fig 1B).

Scratches were manually applied that were several millimeters distant from each other. This form of scratching did not induce any bleeding and resulted only in very minor tissue loss. A small area of consistently 50–200 $\mu m$ from the epidermis was removed superficially, including disruption of the underlying Laminin-332–positive basement membrane but with no other visible damage to the neighboring epidermis and its appendages, including hair follicles (Fig S1A and B). In addition, in contrast to full-thickness wounds, scratch wounding did not cause any substantial damage to the dermis and induces only a local dermal remodeling response as indicated by changes in fibronectin (FN) expression (Fig 1C). This constellation is typical for previously reported partial-thickness wounds (Rittie, 2016). To further characterize these wounds, we performed immunofluorescence staining on E-cad-CFP mice that constitutively express CFP as a fusion protein with E-cadherin, thereby marking epidermal cell–cell contacts (Snippert et al, 2010). Within the wound, both the keratin 14–positive basal layer and the keratin 10–positive suprabasal layer of the epidermis were injured

and disrupted (Fig 1D). Next, we analyzed whether the inflammatory phase of wound closure previously described for full-thickness wounds is also a tissue response upon scratch wounding. To this end, anti-CD45 immunostainings were carried out on tissues obtained immediately, 8, 16, and 24 h after scratch wounding. CD45-positive leukocytes could not be detected directly after wounding but appeared after 8 h post wounding. This observation can be taken as an indication for a rapidly triggered innate immune response as described for full-thickness wounds (Gurtner et al, 2008). Furthermore, 16 h post-injury wounds had developed a fibrin clot covering the wound site, which is also in agreement with descriptions of an immediate wound response of biopsy-induced full-thickness wounds (Fig 1D) (Gurtner et al, 2008).

Taken together, we established a model of superficial scratch wounds in a fluorescent mouse model that clearly differs from full-thickness wounds and that affects the epidermis, while leaving the dermal architecture mostly intact.

### Mode of migration during scratch wound closure

To be able to faithfully determine and distinguish between basal and suprabasal layer upon wounding, we performed scratch wounds in E-cad-CFP mice. We identified the keratin 14–positive basal and keratin 10–positive suprabasal layer by immunofluorescence staining on wounded skin whole mounts (Fig S2A). Using an unbiased machine learning approach, we used E-cad-CFP-signal to segment cells located at the basal and suprabasal layers. This analysis revealed that the more columnar and cuboidal basal keratinocytes are smaller (50–300 $\mu m^2$) than the differentiated polygonal shaped and flattened suprabasal keratinocytes (~300–600 $\mu m^2$) (Fig 2B). In both layers, the size of cells does not significantly change when located at different distances from the wound (Figs 2A and S2B). Moreover, these differences in size between basal and suprabasal cells remain constant over time during wound healing (Fig 2B), illustrating that we can distinguish both layers based on the size of the cells. The constancy of cell sizes over time is in contrast to what was reported previously upon biopsy-induced full-thickness re-epithelialization (Aragona et al, 2017) where a polarized elongation of basal keratinocytes towards the wound bed within the first 2 d post wounding was observed. This may simply reflect the physiological difference between the two types of wounds. Healing of full-thickness wounds is a process of multiple days and re-epithelialization takes places on top of a provisional granulation tissue matrix where keratinocytes need to cover a distance of several millimeters. In our model, the wound has only a size of 50–200 $\mu m$ which corresponds to ~10–40 diameters of basal cells.

Previously, three different modes of wound bed re-epithelization have been proposed based on observations in full-thickness wound repair in vivo and on in vitro scratch assays (Fig 2C) (Omelchenko et al, 2003; Poujade et al, 2007; Safferling et al, 2013; Rittie, 2016). To identify the re-epithelialization strategy of partial-thickness wounds in our murine model, we performed time-lapse intravital microscopy during the entire process of scratch wound closure on the back skin of the mouse. Using the distinct cell size and relative position along the axial axis, we monitored the distinct migration behavior of basal and suprabasal keratinocyte upon tissue repair. We identified that wound closure happens in two steps: (1) keratinocytes start to move

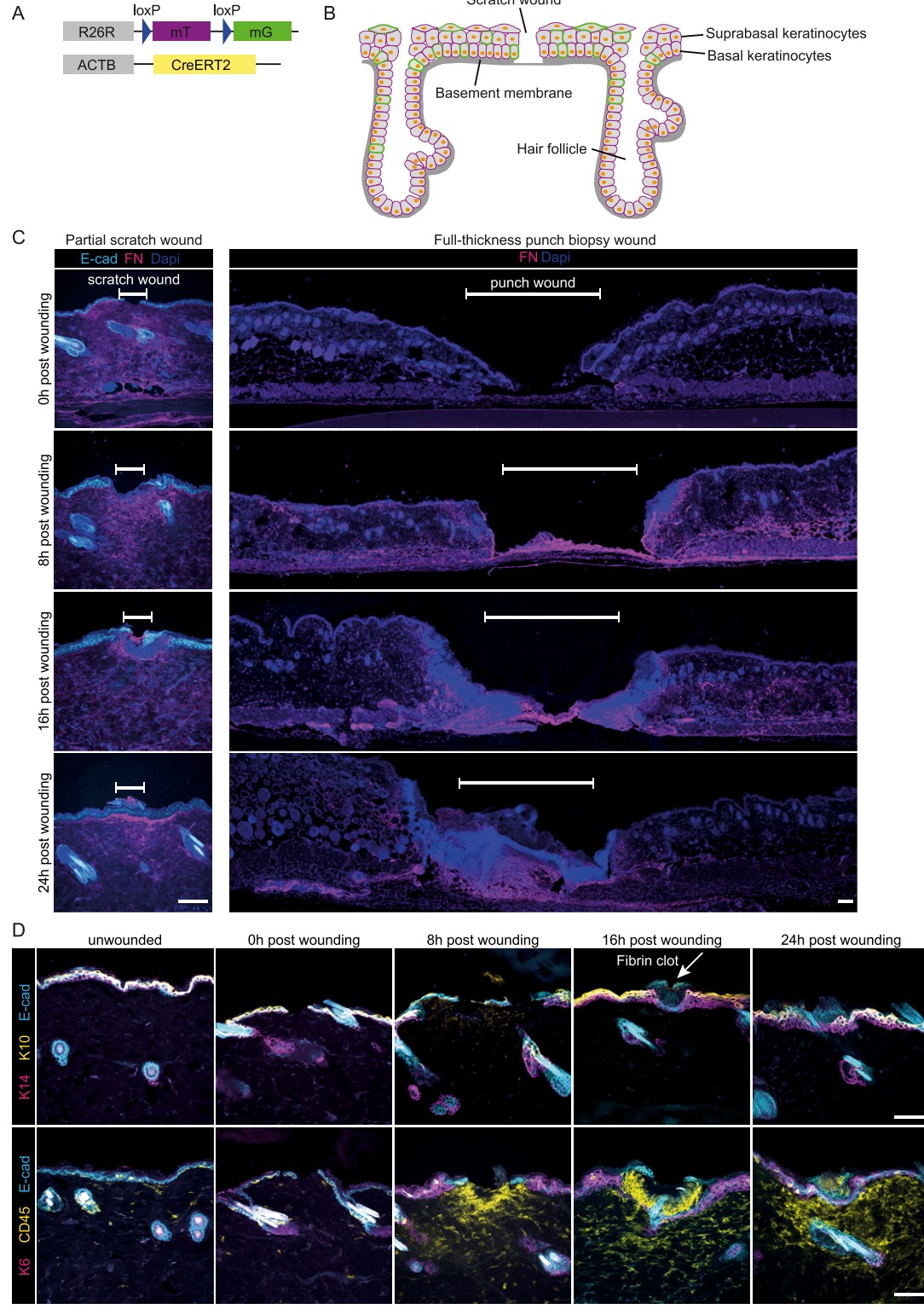

**Figure 1. Murine scratch wounding model follows physiological phases of wound healing.**
**(A)** Mouse model, *R26*-CreERT2:*R26*-mTmG, before Cre recombination cells ubiquitously express a membrane-localized tdTomato. Upon Tamoxifen injection, Cre is activated and randomly and dose-dependently recombines the reporter cassette from tdTomato to GFP. Therefore, some cells start to express membrane-localized GFP. **(B)** Schematic murine epidermis upon scratch wounding. **(C)** Immunofluorescence stainings immediately (0), 8, 16, and 24 h after wounding. Fibronectin (FN, magenta) indicates dermal remodeling in scratch wounds (left panel, scale bar 100 $\mu$m) or full-thickness punch biopsies (right panel, scale bar 100 $\mu$m) of E-cad-CFP mice. White bar indicates the wound site. DAPI, blue. **(D)** Immunofluorescence stainings detecting keratin 14 (basal cell marker, magenta), and keratin 10 (suprabasal cell marker, yellow) (top panel), with arrow pointing to a fibrin clot. Keratin 6 (stress response marker, magenta) and CD45 (leukocyte marker, yellow) (bottom panel) in the skin of E-cad-CFP mice (cyan) (representative images from n = 3 mice). Scale bar, 100 $\mu$m.

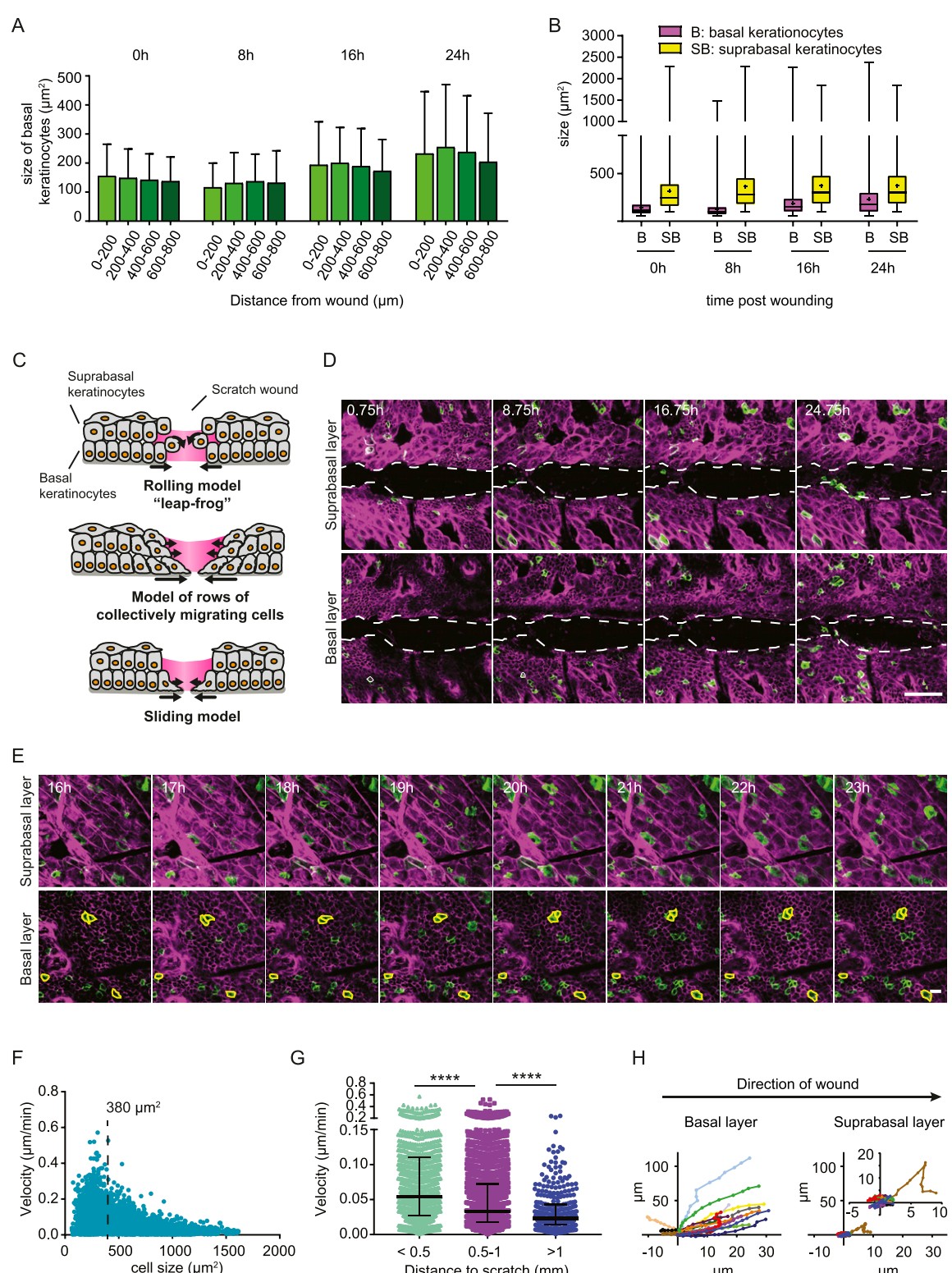

**Figure 2. Basal keratinocytes migrate as a sheet between two static layers.**
**(A)** Measurements of cell areas ($\mu m^2$) of basal cells based on E-cad-CFP expression in different distances towards the wound, directly after wounding (0), 8, 16, and 24 h post wounding. (n = 2 individual mice, with four wounds each per time point). **(B)** Measurements of cell areas of basal (B, magenta) and suprabasal cells (SB, yellow), directly after wounding (0), 8, 16 and 24 h post wounding. (n = 2 individual mice, with four wounds each per time point). **(C)** Schematic representation of three proposed models of modes of migration during re-epithelization. **(D)** Representative sequential images of intravital microscopy of the suprabasal layer (top panel) and the basal layer (bottom panel) after scratch wounding. Scale bar, 100 $\mu m$. See also Video 1. **(E)** Representative images of intravital microscopy of a defined area in proximity (290 $\mu m$

immediately towards the wound but stall at the wound bed, (2) basal cells start entering the wound bed.

Surprisingly, we did not observe any keratinocyte migration into the wound bed within the first 20 h post wounding (n = 3 mice with 3–6 imaged scratch wounds/mouse) (Fig 2D and Video 1). Instead, by tracking the randomly GFP-labelled cells within the layer of mTom+-cells we observed that basal keratinocytes started migrating immediately post wounding as a connected sheet towards the wound that jammed at the wound site. In contrast, we did not observe significant displacement of suprabasal cells (Video 2). 20 h after wounding, basal keratinocytes were the first to migrate into the wound bed and they did this as a continuous cell sheet (Fig 2D). Although the initial push towards the wound bed and subsequent jamming at the wound side of basal keratinocytes might be initiated by the loss of contact inhibition of locomotion, further signals originating from the inflammatory response might be the requirement for subsequent migration towards wound closure. A recent study that followed keratinocyte re-epithelialization upon full-thickness wounding in the ear skin, also observed a delayed activation of migration towards the wound bed (Park et al, 2017). However, in their model, basal as well as suprabasal keratinocytes were migratory, suggesting again profound differences in migration modes between partial- and full-thickness wounds.

To examine the influence of the distance of keratinocytes to the wound edge on their migratory activity, we compared defined areas positioned less than 0.5 mm and up to more than 1 mm (1.0–2.5 mm) away from the wound 16 h after wounding (Fig 2E). Starting at 16 h post wounding and imaging for 8 h would cover both times before and after, respectively, the first basal keratinocytes would enter the wound bed. We determined the migration velocity of all keratinocytes in relation to their cell size, which demonstrated that mostly the smaller (basal) keratinocytes (<380 $\mu m^2$) migrated towards the scratch wound, whereas bigger cells were less migratory (Fig 2F). As reported for the full-thickness wounds (Park et al, 2017), the velocity of migration showed a high spread between individual cells. In addition, we evaluated migration velocity of smaller (basal) keratinocytes in relation to their distance to the wound (Fig 2G). Basal cells closer to the wound edge showed a higher migration velocity than basal cells more distal to the wound site. While cells close to the wound edge (<0.5 mm from scratch) migrated with a median velocity of 0.05 $\mu m$/min, cells that were >1 mm away from the edge migrated half as fast (~0.025 $\mu m$/min), suggesting that leading edge basal keratinocytes are recruited fast to the injured tissue to cover the wound bed once the jammed phase is overcome. To test whether the directionality of migration towards the wound is different for basal and suprabasal cells, we constructed rose plots of representative images (Fig 2H). Whereas most suprabasal keratinocytes migrated little and displayed a non-directional random movement, basal keratinocytes showed prominent directed movement towards the

wound site. These data show that basal keratinocytes actively re-epithelialize the wound bed, while suprabasal keratinocytes are behaving rather passively and remain in place.

Taken together, the finding that suprabasal cells are rather static and do not move toward the wound bed excludes the leap-frog model as a mode of migration during scratch wound healing. In addition, we can also rule out the model of collectively migrating epidermal layers because it is only the basal layer and not the suprabasal layer that moves towards the wound. The sliding model, in which basal keratinocytes at the leading edge migrate towards the wound bed comes closest to our observations.

## Characterization of single cell dynamics during wound closure

The sliding model suggests that basal keratinocytes at the leading edge initiate directed migration towards the wound, whereas neighboring keratinocytes follow passively because of their connection to leader cells within the basal layer (Farooqui & Fenteany, 2005). If the model applies, migration behavior of basal keratinocytes more distal to the wound should be determined by the migration dynamics of leading wound-edge keratinocytes. To test this hypothesis, we analyzed the mode of migration behavior of basal keratinocytes at the single-cell level during their entry into the wound bed. Within a migrating epidermal layer towards the wound, basal cells did not behave uniformly. Instead, each cell migrated individually within the collectively migrating sheet. This becomes particularly apparent when pairs of randomly GFP+-labelled basal cells were tracked (Fig 3A). Therefore, we categorized GFP+-basal cell pairs into those positioned closely (<0.5 mm), intermediately close (0.5–1 mm), or distal (>1.0 mm) from the wound site and tracked their behavior over time. We found that cells constantly lose contact with their original neighbors and establish contacts with new neighboring keratinocytes independent of their position relative to the wound (Fig 3B). Intriguingly, the frequency of exchanging neighboring cells increased relative to the distance to the wound with a higher frequency of exchanging neighbors of cells being closer to the wound bed (Fig 3C). Moreover, faster migrating cells closer to the wound exchanged their neighboring cells more frequently than cells further away from the wound bed (Fig S3). Based on these findings, we can partially reject the model of leading cells that initiate and transmit migratory behavior of a static basal epidermal layer. Instead, we find swarm behavior where cells migrate individually in a collective group towards the wound, such as schooling of fish in the sea.

## Hair follicle obstacles and keratinocyte migration

The observation that basal keratinocytes are able to frequently change neighboring cells and thereby constantly making an effort

away from the wound edge) to the wound bed starting 16 h after wound induction; suprabasal keratinocytes (top panel) and basal keratinocytes (bottom layer). Individual basal keratinocytes highlighted with a yellow outline. Scale bar 100 $\mu m$. See also Video 2. **(F)** Quantification of mean migration velocity in relation to cell size 16 h after induction of scratch wounding. (n = 5 individual mice). Dashed line indicating 380 $\mu m^2$, separating basal from suprabasal keratinocytes. **(G)** Quantification of mean migration velocity of migratory (>10 $\mu m$/8 h) basal keratinocytes in relation to the distance of the wound 16 h after induction of scratch wounding. Median and interquartile distances are plotted for the distance bins <0.5 mm; 0.5–1 mm; >1 mm. (n = 5 individual mice). ****$P$ > 0.0001, Two-tailed Mann–Whitney U-tests were performed for all statistical analyses. **(H)** Rose plot showing migration directionality in respect to the wound site of representative basal (left) and suprabasal cells (right). Inner rose plot in suprabasal cells represents a zoom of the initial rose plot. Value 0 indicates the relative starting point of migration 16 h post wounding. Source data are available for this figure.

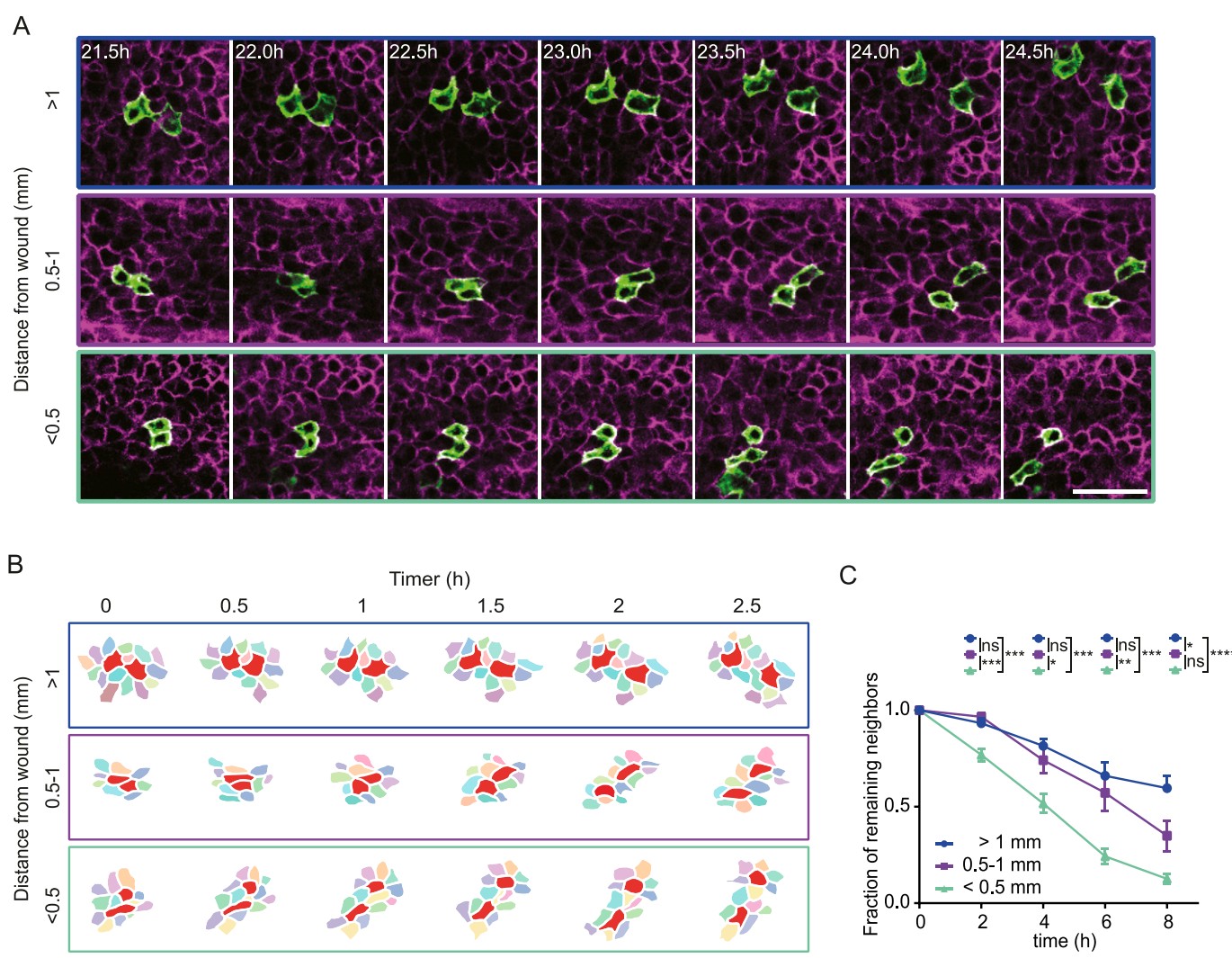

**Figure 3. Single keratinocytes move within the collectively migrating sheet.**
**(A)** Representative images of a migrating pairs at different distances to the scratch of recombined GFP-expressing cells in a R26-CreERT2:R26-mTmG mouse 16 h after scratch wound induction. Scale bar, 50 µm. **(B)** Reconstruction of migrating cell pairs and their neighboring cells, followed over minimal 2 h 30 min, 16 h after scratch wounding. The reconstructions depict three examples of cell group with different distances to the scratch wound. **(C)** The fraction of original neighbors that are remained is plotted over time. Basal keratinocytes are grouped based on initial distance to scratch wound (>1 mm; 0.5–1 mm; <0.5 mm). Imaging was performed 16 h post scratch wounding for 8 h. Plotted is the mean ± SEM for 25 imaging positions in three mice. ANOVA multiple comparison with *$P > 0.1$, **$P > 0.01$, and ****$P > 0.0001$.

to disassemble and re-assemble cell–cell contacts, raised the question whether this migration strategy could be functionally relevant. In contrast to a classical in vitro scratch assay, the undamaged skin surrounding the scratch wound contains intact hair follicles. Keratinocytes moving towards the wound may potentially be blocked by hair follicles. Therefore, we wondered whether the swarm migration behavior of interfollicular basal keratinocytes enabled them to bypass hair follicles. We used intravital microscopy on scratch wounds in fluorescent ubiquitylation-based cell cycle indicator 2 (Fucci2) mice in which the nucleus of cells is fluorescently labelled. We found that keratinocytes bypassed hair follicles without altering overall directionality (Fig 4A and Video 3). Basal keratinocytes on their way towards the wound side that arrived at the point of a hair follicle exchanged their neighbors to circumvent the obstacle and to continue migrating in their direction (Fig 4A).

Our imaging setup also enables us to test whether in scratch wounds, similar to full thickness wounds, stem cells within the upper part of the hair follicle are able to migrate out and contribute to epidermal wound repair (Aragona et al, 2017). Using our mTmG mouse model, we imaged hair follicles in different distances to the wound. We never observed GFP⁺ cells migrating out of a hair follicle. Independent of the hair follicle distance to the wound, GFP⁺ cells within the infundibulum and isthmus upper part of the hair follicle did not migrate at all and stayed attached in the hair follicle, while hair follicle-surrounding basal cells followed their migration track towards the wound, suggesting a hair follicle-independent mode of scratch wound healing (Fig 4B).

Based on the combined data, we propose that only interfollicular epidermal basal keratinocytes migrate individually in a swarming movement towards the wound that enable them to bypass hair follicles.

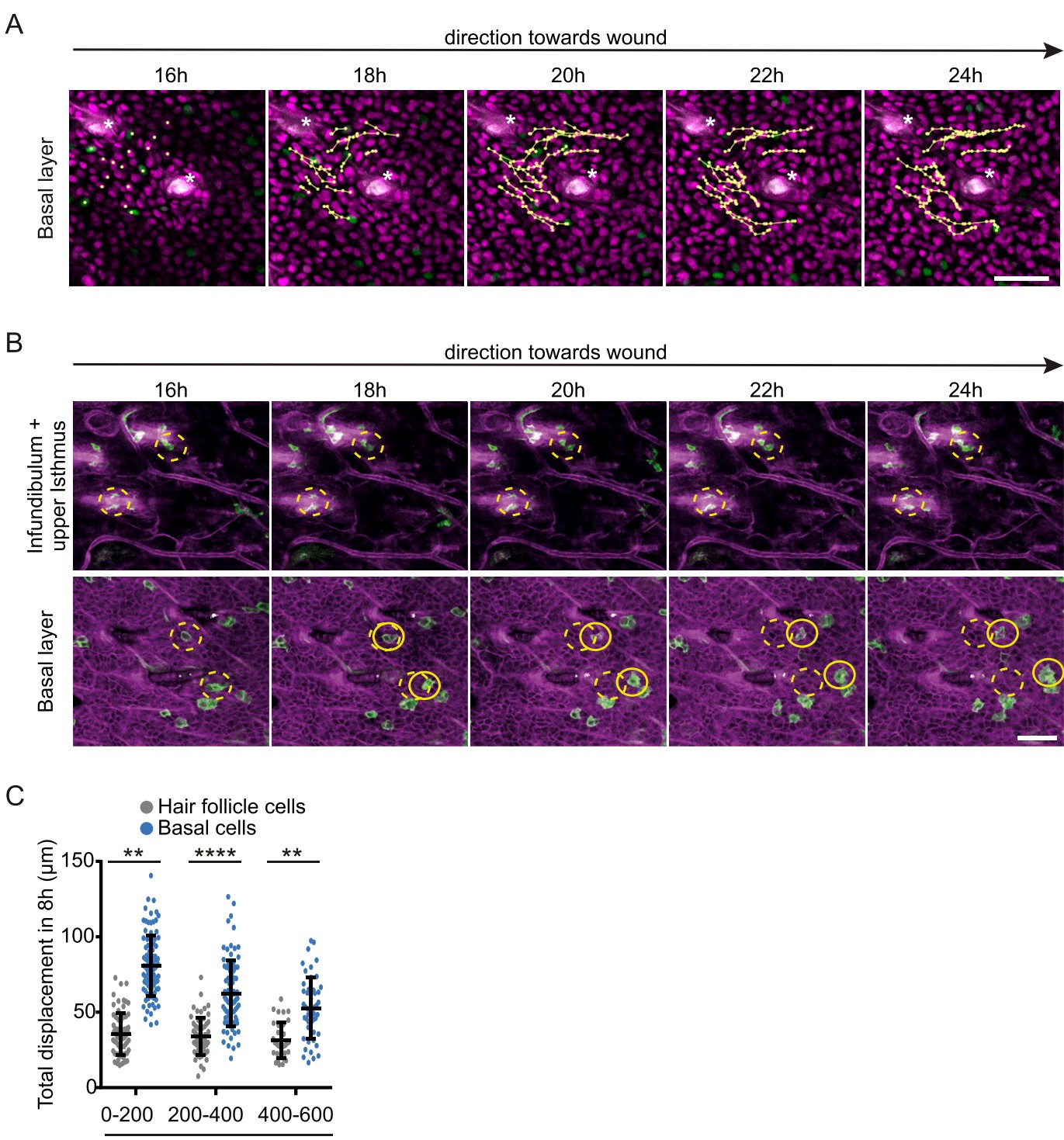

**Figure 4. Migrating keratinocytes bypass hair follicle obstacles.**
**(A)** Representative images of migrating keratinocytes in a Fucci2 mouse 16 h after scratch wound induction bypassing a hair follicle (*). Scale bar, 50 μm. Individual keratinocytes are highlighted by their migration tracks. Also see Video 3. **(B)** Representative sequential images of labelled cells within mTmG mice 16–24 h post wounding within the infundibulum and upper hair follicle (upper panel) and cells within the basal layer (lower panel). Migration/no migration is indicated by the shifts in yellow circles over time. **(C)** Quantification of total displacement in different distances towards the wound of hair follicle (grey) and basal cells (blue), respectively. Data points represent single cells from two mice, with five different wounds per mouse. ****$P < 0.0001$; **$P = 0.002$ (unpaired $t$ test with Welsh correction). Source data are available for this figure.

### Analysis of proliferation and how it interweaves with migration

Immediate keratinocyte migration towards the wound is crucial to close the wound side. However, the loss of cells upon wounding needs to be replenished by proliferation. We identified by 5-ethynyl-2′-deoxyuridine (EdU) incorporation that most proliferation was initiated 16 h post-scratch wound initiation (Fig 5A). To further characterize this proliferation in time and space in detail, we imaged repair of scratch wounds in Fucci2 mice (Abe et al, 2013). In these mice, individual cells can be identified and tracked based on their differential expression of mCherry-hCdt1 (magenta) in cells that are in a G1-cell cycle state and mVenus-hGem (green) in proliferating cells in S/G2 phase (Fig 5B and Video 4). Using intravital microscopy starting 16 h post-scratch wounding, we identified that ~5% of cells proliferate directly next to the wound site (0–200 μm); however, the majority of proliferating cells (~10%) are localized in a zone 200–400 μm away from the wound side (Fig 5B and C). Within this zone the percentage of proliferating cells stays stable during the course of wound healing (Fig 5D). The existence of a proliferative zone has also been shown during wound healing of full-thickness wounds (Aragona et al, 2017; Park et al, 2017). In contrast to the reported proliferative zone in full-thickness wounds, we find for our partial-thickness model that the proliferative zone establishes earlier after wounding and in closer proximity to the wound bed, indicating that induction of wound healing response in time and space might be dependent on the size of wounding. To test this idea, we correlated the width of a scratch to the induction of proliferation and migration at a distance 200–600 μm away from the wound site (see Fig S1B). Indeed, we found a strong correlation between those parameters (Fig 5E and F), suggesting that wound size dictates the number of cells that need to be replenished, and therefore the amount of proliferation and migration velocity of surrounding keratinocytes.

The observed persistence of a proliferative zone suggests that proliferation might be uncoupled from the response of keratinocytes to quickly migrate towards the wound bed with high velocity. To test this, we tracked individual cell in G1- or S/G2-phase, respectively, in the basal layer of the epidermis in different distances towards the wound bed. We found that proliferative (S/G2) and non-proliferative (G1) basal cells migrate with the same velocity (Fig 5G) and directionality (Fig 5H). Therefore, the proliferative state of a cell does not influence its ability to migrate.

### Final remarks

Re-epithelialization of the skin after partial-thickness wounding has not yet been characterized in real time in vivo. Here, we established a partial-thickness wounding model that allowed us to study migration of individual keratinocytes within the native tissue context during re-epithelialization in the back skin of a mouse. Our model allowed us to rigorously test previously described models of re-epithelialization. The observation that mostly basal keratinocytes contribute to re-epithelialization excludes the leap-frog model as a mode of migration, as well as the model that proposes collective migration of directly linked basal and suprabasal cells. The sliding model may apply for the basal keratinocytes at the very front of the basal layer. However, we found that most actively migrating basal keratinocytes migrate individually within the basal layer, such as a swarming movement. This type of movement requires continuous changes in contacts to neighboring keratinocytes and the basement membrane (Collins & Nelson, 2015; Friedl & Mayor, 2017). This high degree of plasticity was quite unexpected. It may, however, reflect the unique properties of the basal cell layer to quickly and on demand activate different adhesion receptors such as integrins or adherence junctions proteins and can therefore ensure fast tissue replacement in case of wounding (Niessen et al, 2011; Wickstrom & Niessen, 2018).

Our findings also exclude the previously proposed model that the leading edge directly transmits forces to adjacent keratinocytes more distant from the wound bed to physically drag them along. Therefore, we suggest a model in which leading edge keratinocytes are the drivers of wound closure and while these cells start to occupy the small wound bed space, individually moving keratinocytes within a cohesive layer follow. Since all keratinocytes in the basal layer migrate towards the wound, the movement should be considered to be collective, despite the continuous change in relative individual cell position (Fig 6). These findings are in line with the studies on plithotaxis migration observed in epithelial monolayers (Tambe et al, 2011; Trepat & Fredberg, 2011); however, further studies are required to elucidate local cellular stress changes upon re-epithelialization in detail. We show for the first time that a patterned collective behavior occurs in vivo during re-epithelialization of superficial wounds providing the degree of plasticity not only to efficiently and rapidly cover differently shaped superficial wounds of different diameters but also to ensure that cells reach the wound even in the presence of obstacles.

## Materials and Methods

### Experimental model and subject details

All mice were from mixed Bl6/FVB background and housed under standard laboratory conditions receiving food and water ad libitum. All experiments were approved and performed according to the guidelines of the Animal Welfare Committee of the Royal Netherlands Academy for Arts and Sciences and The Netherlands Cancer Institute, The Netherlands. R26-mTmG:R26-CreERT2 mice (Muzumdar et al, 2007; Ventura et al, 2007) were used to label and trace single cells, Fucci2 (Abe et al, 2013) mice were used to analyze proliferation and E-cad-CFP mice (Snippert et al, 2010) were used to determine wound characteristics and cell sizes. At the age of 8- to 11-wk mice were enrolled in the experiments. 48 h before imaging, R26-mTmG:R26-CreERT2 mice were intraperitoneally injected with tamoxifen (0.2 mg/25 g; Sigma-Aldrich; diluted in sunflower oil) to activate Cre recombinase. 24 h before the initiation of wounds, the hair from the back skin was removed using hair removal cream (Veet). Scratch wounds were induced manually using a 26G needle on the hairless back. Full-thickness punch biopsies (2 mm) were set in the middle of the hairless back skin.

### Intravital microscopy

For imaging, mice were sedated using isoflurane inhalation anesthesia (~2.0% isoflurane/compressed air mixture) and received

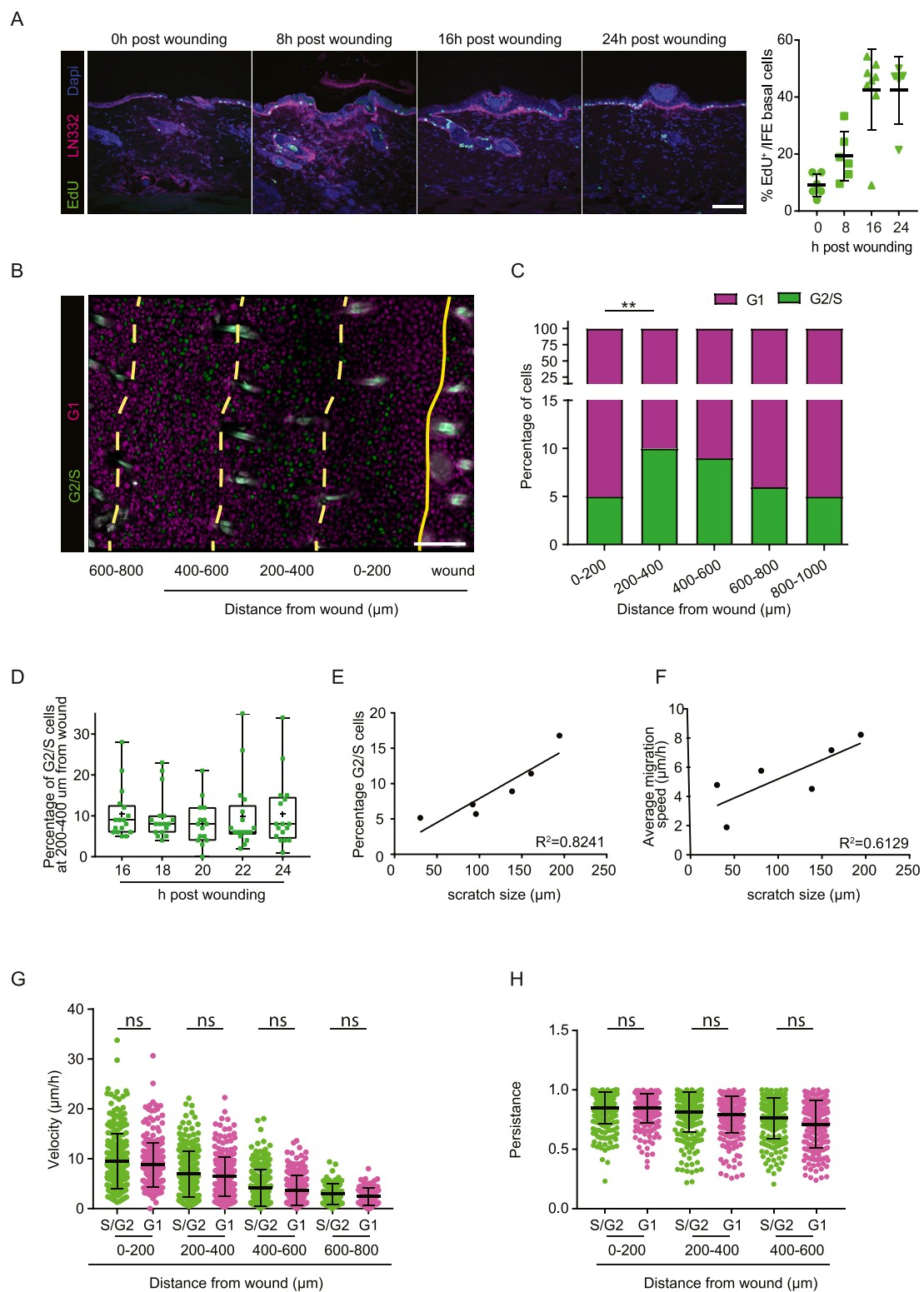

**Figure 5. Proliferation occurs in a distinct zone and does not affect overall migration.**
**(A)** Left panel: fluorescence staining of EdU incorporation (green) immediately after wounding and 8, 16 and 24 h post wounding. LN332 (magenta) immunofluorescence counterstain marks the basement membrane. Scale bar 100 $\mu$m. Right panel: quantification of EdU incorporation, each dot represents one wound in n = 2 mice.
**(B)** Representative image of a scratch wound in Fucci2 mice. Cells in G1-phase (magenta), in G2/S-phase (green) in their different distances towards the wound. Scale bar 100 $\mu$m. **(C)** Quantification of percentage of proliferating (G2/S, green) and non-proliferating cells (G1, magenta) grouped based on their distance to the wound (0–200, 200–400, 400–600, 600–800, and 800–1,000 $\mu$m). **$P = 0.0086 (ANOVA multiple comparison). **(D)** Percentage of proliferating G2/S cells within a distance of 200–400 $\mu$m

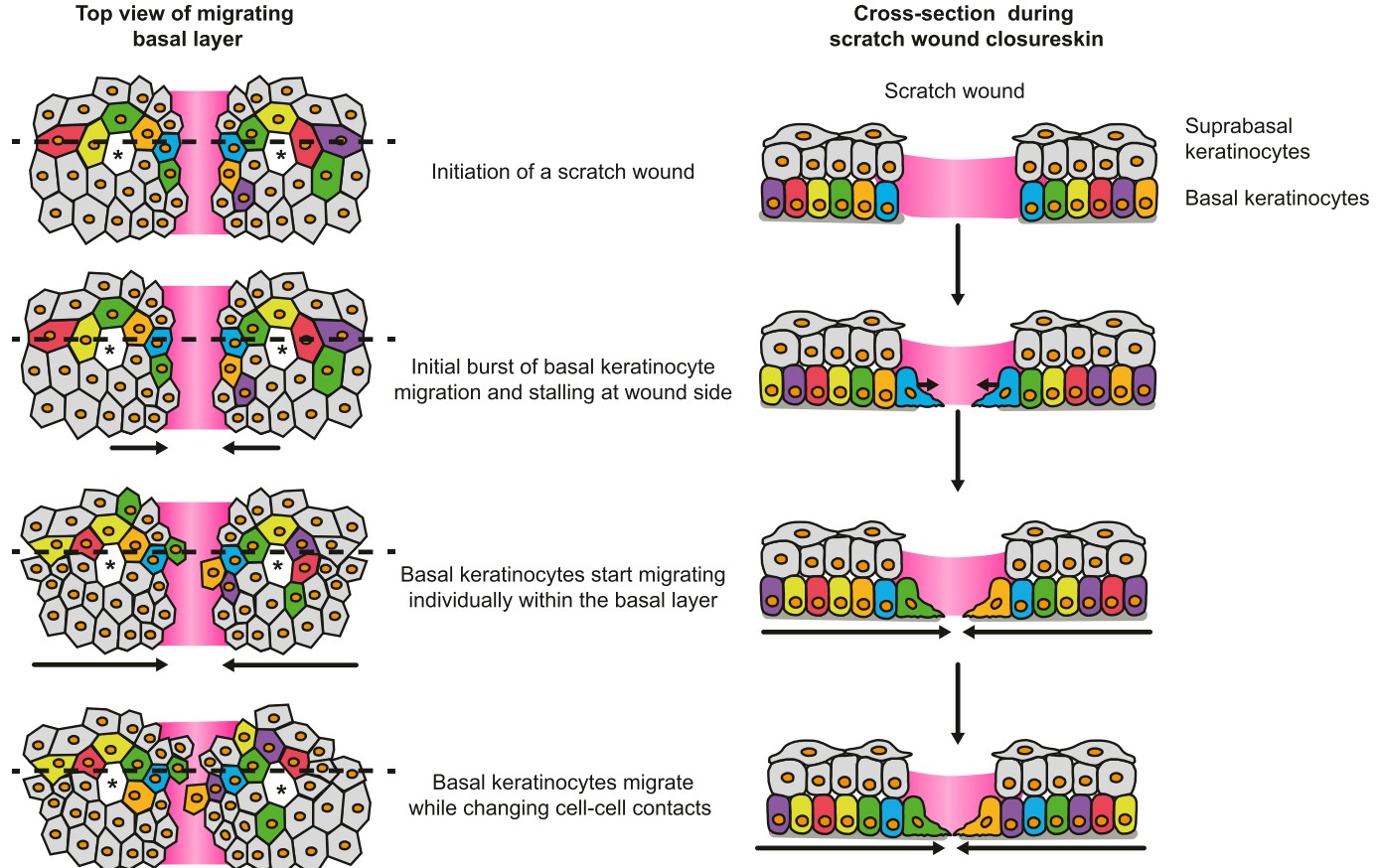

**Top view of migrating basal layer**

**Cross-section during scratch wound closureskin**

Scratch wound

Suprabasal keratinocytes

Basal keratinocytes

Initiation of a scratch wound

Initial burst of basal keratinocyte migration and stalling at wound side

Basal keratinocytes start migrating individually within the basal layer

Basal keratinocytes migrate while changing cell-cell contacts

**Figure 6. Model of scratch wound re-epithelialization.**
Schematic representation of the model for re-epithelization in partial-thickness wounds. Upon the initiation of a scratch wound, basal keratinocytes immediately move towards the wound side without entering the wound bed. After a period of jamming, basal keratinocytes enter the wound bed, whereas suprabasal keratinocytes remain static. Basal keratinocytes migrate individually through a collectively moving basal layer directed towards the wound bed. Left panel, top view of the basal migrating keratinocytes. Dashed line representing the cross section shown in the right panel.

200 µl sterile PBS by subcutaneous injection. Mice were then placed in a custom-designed imaging box on the microscope stage and were kept under constant anesthesia with the imaging box and the microscope adjusted to 34.5°C using a climate chamber. For imaging sessions longer than 8 h, mice were supplied with glucose and electrolytes via a subcutaneous infusion (NutriFlex special 70/240, 100 µl/h; Braun). Intravital microscopy was performed using an inverted Leica TCS SP5 confocal microscope equipped with an argon laser and a DPSS 561-nm laser or using an inverted Leica SP8 Dive system with a MaiTai eHP DeepSee laser (Spectra-Physics) and Insight (Spectra-Physics). Using confocal microscopy, fluorophores were excited as follows: GFP at 488 nm and mTomato at 561 nm. GFP fluorescence was collected at 492–530 nm and mTomato fluorescence at 580–640 nm. The multiphoton system was equipped with four HyD-RLD detectors;

GFP and mTomato or mVenus and mCherry were simultaneously excited with 960 nm (Insight X3) and singles, including second harmonic generation (Collagen I, stroma) were detected using the hybrid detector. Images were recorded every 30 min as three-dimensional tile scans with 5-µm Z-step size at 12 bit. They were acquired with a 25× water immersion objective with a free working distance of 2.40 mm (HC FLUOTAR L 25×/0.95 W VISIR 0.17).

## Post-processing and analysis of intravital microscopy data

For analysis, three-dimensional tile scan videos were corrected for XYZ drift using custom-made software (codes on request available from R Windoffer/J van Rheenen) and processed using Fiji. Migration was tracked using Amira and costume-made MatLab software (available

away from the wound bed is plotted between 16 and 24 h post wounding. Plots depict the min., max., and median, the lower and upper quantile, the plus indicates the mean. **(E)** Linear regression analysis between scratch size (µm) and percentage of G2/S-phase proliferating cells 200–600 µm away from the wound site. $R^2$ = 0.8241. **(F)** Linear regression analysis between scratch size (µm) and average migration velocity (µm/h) of basal keratinocytes 0–200 µm away from the wound site. $R^2$ = 0.6129. **(G)** Quantification of migration velocity of proliferating S/G2-phase cells (green) and non-proliferating G1-phase cells (magenta) within different distances towards the wound. ns, not significant (unpaired $t$ test). **(H)** Quantification of persistence of migratory proliferating S/G2-phase cells (green) and non-proliferating G1-phase cells (magenta) within different distances towards the wound. ns, not significant (Kruskal–Wallis test).
Source data are available for this figure.

on request from R Windoffer). Migration of Fucci2 mice or of mTmG mice in hair follicles was analyzed using the MTrack2 plugin in ImageJ (Stuurman N, Schindelin J, Elliot E, and Hiner M, http://imagej.net/MTrack2). Persistence of migratory cells (velocity > 5 $\mu$m/h) was determined by deviation of total track length by total displacement.

## Immunostaining

Wounded skin samples were fixed in periodate-lysine-4% paraformaldehyde (PLP) buffer overnight at 4°C, incubated in 30% sucrose overnight at 4°C and embedded in tissue freezing medium (Leica Biosystems). The skin was cryosectioned, and immunostainings were performed on 10-$\mu$m sections. For this, the sections were hydrated in PBS for 10 min at RT and subsequentially blocked and permeabilized for 1 h with 0.5% Triton X/5% NGS in PBS. The following primary antibodies were used: anti-CD45-Alexa Fluor 488 (103122; BioLegend), anti-Keratin6a (905701; BioLegend), anti-K14 (PRB155P; Covance), anti-K10 (GP-K10; Progen), anti-Fibronectin (23750; Abcam), and anti-Laminin 332 (kind gift from M Aumailley, University of Cologne). These antibodies were diluted in 0.1% Triton X/5% NGS in PBS and stained overnight at 4°C. After three times washing in PBS, secondary antibodies (donkey anti-rabbit AF568 [A10042; Invitrogen], goat anti-rabbit AF594 [A11012; Invitrogen], and goat anti-guinea pig AF488 [A11073; Invitrogen]) were incubated for 1 h at RT and nuclei were stained with 4′,6-diamidino-2-phenylindole (DAPI, 0.1 $\mu$g/ml; Sigma-Aldrich) or TO-PRO-3 (T3605; Invitrogen). After three 10-min washings in PBS, the stained sections were mounted using VectaShield. Staining on punch biopsies was performed on paraffin-embedded wounds. Paraffin sections were deparaffinized and hot target retrieval was performed in citrate buffer, pH 6. Stainings, without permeabilization, were performed as described above. All stainings were imaged with an inverted Leica TCS SP5 and TCS SP8 confocal microscopes. Different fluorophores were excited as follows: DAPI at 405 nm, GFP at 488 nm, Tomato at 561 nm, and Alexa 647 at 633 nm. DAPI was collected at 440–470 nm, GFP at 492–530 nm, Tomato at 570–640 nm, and Alexa 647 at 650–700 nm. All images were collected in 12 bit with 25× water immersion objective (HC FLUOTAR L N.A. 0.95 W VISIR 0.17 FWD 2.4 mm).

## Skin whole mount stainings

Wound samples were isolated, and subcutaneous fat was removed by scraping. Tissues were fixed in periodate-lysine-4% paraformaldehyde (PLP) buffer overnight at 4°C. Whole skin samples were permeabilized and blocked for 3 h in 5% BSA/5% NGS/1% Triton X in PBS and subsequently stained with primary antibodies (anti-K14 [PRB155P; Covance] and anti-K10 [GP-K10; Progen]) in blocking solution overnight at RT. After three washes with 0.2% Tween-20 in PBS, secondary antibodies (goat anti-guinea pig AF488 [A11073; Invitrogen] and chicken anti-rabbit AF647 [A21443; Invitrogen]) were incubated for 6 h at RT. Skin samples were washed three times with 0.2% Tween-20 in PBS and mounted in VectaShield Hard Set. All whole mounts were imaged with an inverted Leica SP8 Dive system with an Insight X3 laser (Spectra-Physics). Different fluorophores were excited as follows: CFP with MP 840 nm, AF 488 at 488 nm, and Alexa 647 at 633 nm. CFP was collected at 430–470 nm, AF 488 at 492–530 nm, and AF 647 at 640–700 nm. All images were collected in 12 bit with 25× water immersion objective (HC FLUOTAR L N.A. 0.95 W VISIR 0.17 FWD 2.4 mm).

## EdU detection

5-Ethynyl-2′-deoxyuridine (EdU) was i.p.-injected 4 h prior euthanasia (1 mg per animal, 5 mg/ml stock in PBS; 900584; Sigma-Aldrich). EdU staining was performed on 10 $\mu$m thick cryo sections by incubating with 100 mM Tris, pH 8.5, 1 mM CuSO$_4$, 100 mM ascorbic acid, and 10 $\mu$M Alexa Fluor 488-Azide (A10266; Invitrogen) for 30 min at room temperature. Quantification was performed manually by counting the number of EdU-positive cells per total basal keratinocytes in the interfollicular epidermis.

## Automated membrane detection and cell area measurements

Fluorescence images of E-cad-CFP of the K14-positive layer or the K10-positive suprabasal layer were subjected to segmentation. Segmentation was performed using the ImageJ plugin Labkit which automatically segments for fore- and background on a trained classifier. This segmentation was used in ImageJ to analyze particles with a circularity of 0.2–1 and a size between 60 and 2,500.

## Statistical analysis

Statistical analyses were performed in Prism (GraphPad). Two-tailed Mann–Whitney U tests and a two-tailed unpaired $t$ test with normality test were used, with $P$-values being reported in the figure legend. One-way ANOVA was used for multiple comparisons. Kruskal–Wallis test was used for multiple comparison of non-normal distributed data. For all box plots, the center-line represents the median, cross represents mean, box limits represent upper and lower quantiles, and whiskers represent minimum and maximum values.

# Supplementary Information

# Acknowledgements

The authors would like to thank the staff at the Netherlands Cancer Institute and Hubrecht Institute animal facility and Bioimaging facility for technical support and input, and members of the InCEM consortium, Nadieh Kuijpers, Nikos Kavalopoulos, and the van Rheenen laboratory for discussions. This work was supported by an European Union's Horizon 2020 research and innovation program under the Marie Sklodowska-Curie grant agreement No 642866 (to L Bornes, R Windoffer, RE Leube and J van Rheenen) an European Research Council CoG Cancer-recurrence (648804), Cancer Genomics Netherlands, and Doctor Josef Steiner Foundation (to J van Rheenen) and European Molecular Biology Organization (ALTF 202-2016 to J Morgner).

## Author Contributions

L Bornes: conceptualization, resources, data curation, formal analysis, funding acquisition, investigation, visualization, methodology, and writing—original draft, review, and editing.
R Windoffer: software, formal analysis, funding acquisition, and writing—review and editing.
RE Leube: resources, funding acquisition, and writing—review and editing.

J Morgner: conceptualization, data curation, formal analysis, supervision, funding acquisition, investigation, visualization, methodology, and writing—original draft, review, and editing.
J van Rheenen: conceptualization, supervision, funding acquisition, project administration, and writing—original draft, review, and editing.

## Conflict of Interest Statement

The authors declare that they have no conflict of interest.

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
