## [Reviewer comments · Life Science Alliance]

Life Science Alliance

Scratch-induced partial wounds re-epithelialize by sheets of independently migrating keratinocytes

Jacco van Rheenen, Laura Bornes, Reinhard Windoffer, Rudolf Leube, and Jessica Morgner
DOI: <https://doi.org/10.26508/lsa.202000765>

Corresponding author(s): Jacco van Rheenen, Oncode Institute, The Netherlands Cancer Institute and Jessica Morgner, Oncode Institute, The Netherlands Cancer Institute

Review Timeline:

Submission Date:	2020-05-05
Editorial Decision:	2020-06-29
Revision Received:	2020-10-26
Editorial Decision:	2020-11-15
Revision Received:	2020-11-17
Accepted:	2020-11-18

Scientific Editor: Shachi Bhatt

Transaction Report:

June 29, 2020

Re: Life Science Alliance manuscript #LSA-2020-00765-T

Prof. Jacco van Rheenen
Onco Institute, The Netherlands Cancer Institute
Molecular Pathology
Plesmanlaan 121
Amsterdam, Choose a state 3584 CT
Netherlands

Dear Dr. van Rheenen,

Thank you for submitting your manuscript entitled "In vivo scratch wounds re-epithelialize by sheets of independently migrating keratinocytes" to Life Science Alliance. The manuscript was assessed by expert reviewers, whose comments are appended to this letter.

As you will see, while the reviewers appreciate the interesting findings presented in your manuscript, they also raise some points to be addressed and have some suggestions that could further strengthen the work. Given the overall high level of interest in your study, we would like to invite you to submit a revised version.

While each of the specific points that were raised should be addressed, it is possible that some of the points raised by Reviewer 2 may be outside the scope of the current study. We would be happy to discuss individual revision points further with you should this be helpful.

In our view these revisions should typically be achievable in around 3 months. However, we are aware that many laboratories cannot function fully during the current COVID-19/SARS-CoV-2 pandemic and therefore encourage you to take the time necessary to revise the manuscript to the extent requested above. We will extend our 'scooping protection policy' to the full revision period required. If you do see another paper with related content published elsewhere, nonetheless contact me immediately so that we can discuss the best way to proceed.

When submitting the revision, please include a letter addressing the reviewers' comments point by

point.

Thank you for this interesting contribution to Life Science Alliance. We are looking forward to receiving your revised manuscript.

Sincerely,

Reilly Lorenz
Editorial Office Life Science Alliance
Meyerhofstr. 1
69117 Heidelberg, Germany
t +49 6221 8891 414
e contact@life-science-alliance.org
www.life-science-alliance.org

B. MANUSCRIPT ORGANIZATION AND FORMATTING:

Reviewer #1 (Comments to the Authors (Required)):

This is a well executed manuscript which very nicely describes re-epithelization following low level wound induction assessed using intravital microscopy. This MS is in line with this journal requisite aligning very well for publication following some minor additions/edits and is of clear value to the community.

Major comment

Throughout the MS the author clearly define the difference between this work and that of other re full wound versus partial wound - this is a key difference to this MS and its findings and as such NEEDS to be in the title.

This will also help the community look to this work for differences in leap-frog, collective movement etc models for small wound versus punch biopsy full wound for example and will reduce the danger that this work gets confused with district type of full wound work.

e.g.
In vivo scratch wounds re-epithelialize by sheets of independently migrating keratinocytes in partial- versus full-wound healing models

or something similar.

minor comment edits.

for fig 1C please show a zoom for pre wound for completeness - this is in supplemental fig but will help the reader..this is not needed for other figs but will help set the scene.

please add more detail for page 8 re provide a full explanation to non expert readers for the definition for directional movement and persistence. work, how was this assess, defined and explain why > .04 versus .6 are used to defined this?

pg 9 please fix typo...second paragraph..lines 6/7 starting with "Within an migrating

change
within a migrating epidermaland did not behave uniform(ly)

page 10 the author discuss cell swarming...there are lots of refs for this phenomenon/ subject
please add rev on this

I may have missed this but fig 4 B does not get describe ie the model please ref to this more in final remarks

Reviewer #2 (Comments to the Authors (Required)):

This manuscript establishes an *in vivo* scratch wound model to study re-epithelialization using state-of-the-art intravital live imaging. The manuscript argues that basal keratinocytes move independently towards the injury site within a collectively migrating sheet, by continuous disassembly and re-establishment of cell-cell contacts with surrounding cells. In general, the methodology is impressive however the manuscript suffers from overinterpretation of the rather limited data presented.

Major comments:

The authors argue that the scratch wound injury model used here is different from commonly used full-thickness wounds. However, they observe both immune cell infiltration and fibrin clots, suggesting a dermal response in line with full-thickness wounds. A careful investigation of the dermal compartment and response would be needed to argue that this model is distinct from full-thickness wounds.

Several models for re-epithelialization have been proposed, commonly linked to the type of injury model used. It would be interesting to compare re-epithelialization in small and large scratch wounds to see if size, rather than dermal involvement, influences epithelial response.

The study uses a generic actin-CreERT2 mouse model to label keratinocytes, excluding the possibility to label basal or suprabasal cells selectively. The conclusions regarding specific basal/suprabasal responses would be strengthened if authors would look at each cell population independently.

Previous *in vitro* scratch wound assays establish proliferation as a major factor in wound healing. The authors do not comment on proliferation in their work, but it would be relevant to investigate if proliferation and migration correspond or influence each other. The material presented in the manuscript would allow for an investigation of proliferation by correlating clone size and migration patterns.

The authors argue that re-epithelialization after scratch wound is independent of hair follicle stem cell contribution. However, supporting movie 3 shows what appears to be a labelled cell appearing from a hair follicle integrating into the basal layer. More data would be needed to support the claim that there is no HFSC contribution to epidermal wound healing in this model.

The authors claim that there are no alterations in cell size observed during scratch wound healing. No data is presented to support this claim, which would be an important point to establish since all conclusions are based on cell size and localization instead of established basal/suprabasal markers.

Minor points:

Figure 1 -

The authors argue that there is no dermal injury in this model, however in figure 1C it looks like there is dermal reorganization at 8 and 16 h after injury. Could dermal markers be used to confirm or dismiss?

It looks like Krt6 is uniformly high in the epidermis directly after injury. Krt6 is normally expressed upon stress or injury, but undetectable in normal epidermis. Please clarify.

Figure 2-

In figure 2C it would be nice to show how close to the wound bed the images are taken, especially since authors argue that the migration pattern is depending on the distance to the wound. It is also not clear why these specific time points are chosen.

In figure 2D authors display cell size and migratory velocity. Based on the previous publications and wound models (Park et al 2017), it is not impossible that basal and/or suprabasal cells alter the cell size when becoming migratory. This would skew the analysis - perhaps authors could compare the size of migratory and non-migratory cells to rule this out.

It is not entirely clear what figure 2F represents - how is directionality measured, and what would be the rationale for suprabasal cells to have such a strong trend against migrating towards the wound site? How many cells were assessed for directionality of each mouse?

The leap-frog model suggests that suprabasal cells can move down and fill up the basal layer, and authors argue that this does not happen in this injury model. How can authors exclude this possibility with no layer specific labelling? Size is used as the only determinant of cell state, and it is possible that cell size will depend on location within the epidermis.

Figure 3-

Here the authors follow labelled pairs to determine migration trajectories of cell originating in close proximity. The existence of pairs indicates that labelled cells have proliferated after tamoxifen administration (see previous comments on proliferation). Figure 3C indicates that labelled pairs closer to the wound are more likely to separate from each other over time - is this significant compared to the other cell populations measured? And does increase reorganization correlate to proliferation patterns?

In Supplemental figure 2 authors show that the speed of migration is dependent on distance to wound bed - is this change statistically significant? If migration speed is higher closer to the wound, labelled cells would be expected to be further away from each other compared to cells further away from the wound.

Movie 1: seems to run for up to 40h instead of the 20h stated in the figure legend.

Reviewer #3 (Comments to the Authors (Required)):

The authors develop an in vivo partial thickness skin wounding model in mice, in an attempt to provide the first real-time data as to the cellular mechanism of closure. The manuscript does a good job motivating why this is interesting, contrasting with the known biology of full thickness wounds and presenting the imaging data. I think there is a significant contribution here in terms of demonstrating the relatively higher study of basal keratinocytes to motility and the high degree of fluidity and neighbor exchanges among the basal keratinocytes, especially those moving the fastest, closest to the wound edge. I think this study will provide a useful method and new conceptual framework for the field. I have one substantive criticism: the authors state that this fluidity enables them to bypass immobile structures (e.g. hair follicles) and I agree that it could. But I didn't see where in their figures they observed such a fluid migration around an immobile structure. Accordingly, it should not be presented as a conclusion but rather a speculation or a possibility for future investigation.

Minor:

1. Abstract: "behave rather passively" should be either "are rather passive" or "behave passively".
2. Page 9: "behave uniform" should be "behave uniformly".
3. There is something wrong with the axis labels in many of the graphs. When printed, many letters were not visible and the labels could not be interpreted.

Reviewer #1 (Comments to the Authors (Required)):

This is a well executed manuscript which very nicely describes re-epithelization following low level wound induction assessed using intravital microscopy. This MS is in line with this journal requisite aligning very well for publication following some minor additions/edits and is of clear value to the community.

Reply:

We thank this reviewer for his/her time to review our manuscript, and are happy with his/her support to publish our manuscript in Life Science Alliance.

Major comment

1) Throughout the MS the author clearly define the difference between this work and that of other re full wound versus partial wound - this is a key difference to this MS and its findings and as such NEEDS to be in the title.

This will also help the community look to this work for differences in leap-frog, collective movement etc models for small wound versus punch biopsy full wound for example and will reduce the danger that this work gets confused with district type of full wound work.

e.g.

In vivo scratch wounds re-epithelialize by sheets of independently migrating keratinocytes in partial- versus full-wound healing models or something similar.

Reply:

We completely agree with this reviewer and would like to thank for the suggested title. Unfortunately, the journals guidelines explicitly state that the titles have a limit of 100 characters (incl spaces) and the suggested title contains 136 characters.

To point out to the readers that our work concerns partial wounds and not full wounds as in other studies, we have changed the title to:

“Scratch-induced partial skin wounds re-epithelialize by sheets of independently migrating keratinocytes”

This title contains 103 characters. We hope that the editor allows us to use this much more informative title.

minor comment edits.

1) for fig 1C please show a zoom for pre wound for completeness - this is in supplemental fig but will help the reader..this is not needed for other figs but will help set the scene.

Reply:

We thank the reviewer for pointing this out. We repeated the wound healing experiments and performed new stainings to also include representative zoom images of non-wounded skin. These can now be found in **Figure 1D** in the revised version of the manuscript.

2) please add more detail for page 8 re provide a full explanation to non expert readers for the definition for directional movement and persistence. work, how was this assess, defined and explain why $> .04$ versus $.6$ are used to defined this?

Reply:

In our initial manuscript, we aimed to use persistence and directionality measurements to illustrate that although some cells move a bit in the suprabasal layer, that this movement is not directed towards the wound. We agree that non-expert readers that do not fully grasp the concept of persistence measurement may miss this important point. In the revised manuscript, we have reworded this argument and illustrate this point by showing the differential patterns of movement in rose plots of basal and suprabasal cells (**Figure 2G** of the revised manuscript).

At page 8 and 9 of the revised manuscript, the text now reads: *“To test whether the directionality of migration towards the wound is different for basal and suprabasal cells, we constructed rose plots of representative positions (Fig 2H). While the majority of suprabasal keratinocytes migrated little and displayed a non-directional random movement, basal keratinocytes showed prominent directed movement towards the wound site.”*

3) pg 9 please fix typo...second paragraph..lines 6/7 starting with "Within an migrating change within a migrating epidermaland did not behave uniform(ly)

Reply:

We corrected this typo in the revised manuscript.

4) page 10 the author discuss cell swarming....there are lots of refs for this phenomenon/ subject please add rev on this

Reply:

In the revised manuscript, we have added the following two references to discuss cell swarming:

- 1) Collins C, Nelson WJ (2015) Running with neighbors: coordinating cell migration and cell-cell adhesion. *Current opinion in cell biology* 36: 62-70
- 2) Friedl P, Mayor R (2017) Tuning Collective Cell Migration by Cell-Cell Junction Regulation. *Cold Spring Harbor perspectives in biology* 9

4) I may have missed this but fig 4 B does not get describe ie the model please ref to this more in final remarks

Reply:

We thank the reviewer for pointing this out. We refer to the model (which is now **Figure 6**) in the final remarks of the revised manuscript (Page 13, *“Therefore, we suggest a model in which leading edge keratinocytes are the drivers of wound closure and while these cells start to occupy the small wound bed space, individually moving keratinocytes within a cohesive layer follow. Since all keratinocytes in the basal layer migrate towards the wound, the movement*

should be considered to be collective, despite the continuous change in relative individual cell position (Fig 6).”).

Reviewer #2 (Comments to the Authors (Required)):

Review comments Life Science Alliance LSA-2020-00765-T

This manuscript establishes an in vivo scratch wound model to study re-epithelialization using state-of-the-art intravital live imaging. The manuscript argues that basal keratinocytes move independently towards the injury site within a collectively migrating sheet, by continuous disassembly and re-establishment of cell-cell contacts with surrounding cells. In general, the methodology is impressive however the manuscript suffers from overinterpretation of the rather limited data presented.

Reply:

We would like to also thank this reviewer for his/her time to review our manuscript and for the constructive feedback on our manuscript. Below we explain how we have addressed the various concerns of this reviewer.

Major comments:

1) The authors argue that the scratch wound injury model used here is different from commonly used full-thickness wounds. However, they observe both immune cell infiltration and fibrin clots, suggesting a dermal response in line with full-thickness wounds. A careful investigation of the dermal compartment and response would be needed to argue that this model is distinct from full-thickness wounds.

Reply:

We thank the reviewer for this suggestion. In the revised manuscript in **Figure 1C**, we now included images side-by-side of partial scratch wounds and full-thickness wounds to illustrate the differences between those types of wounds. For the full-thickness images, we performed small 2 mm punch biopsies on the mouse back skin and followed dermal remodeling in the same time regime as we follow healing of scratch wounds. We use changes in fibronectin expression as a read-out of dermal remodeling. Unlike punch biopsy wounds, scratch wounds close during the observed 24 hours time-window and show a narrower zone of fibronectin remodeling around the wound margin, both indicating that our scratch wound model is distinct from full-thickness punch wounds.

To illustrate these differences to the reader, we describe this data at page 5 of the revised manuscript: *“Scratches were manually applied that were several millimeters distant from each other. This form of scratching did not induce any bleeding and resulted only in very minor tissue loss. A small area of consistently 50-200 μ m from the epidermis was removed superficially, including disruption of the underlying Laminin-332-positive basement membrane but with no other visible damage to the neighboring epidermis and its appendages including hair follicles (supplementary Fig S1A, B). Additionally, in contrast to full-thickness wounds, scratch wounding did not cause any substantial damage to the dermis and induces only a local dermal remodeling response as indicated by changes in fibronectin (FN) expression (Fig 1C).”*

2) Several models for re-epithelialization have been proposed, commonly linked to the type of injury model used. It would be interesting to compare re-epithelialization in small and large scratch wounds to see if size, rather than dermal involvement, influences epithelial response.

Reply:

We agree that it is interesting to compare re-epithelization of small and large scratch wounds. Due to the method of scratch induction, we obtain small variations in the scratch width (see **supplementary Fig S1B** of the revised manuscript). We used this variation to test a potential correlation between the width of a scratch and induction of proliferation and migration at a distance 200-600 μm away from the wound site.

Interestingly, we indeed see a correlation between wound size and amount of proliferation and velocity of migrating keratinocytes. In the revised manuscript we included these findings in **Figure 5 E** and **F**, and describe them at page 12: *“To test this idea, we correlated the width of a scratch to the induction of proliferation and migration at a distance 200-600 μm away from the wound site (see supplementary Fig S1B). Indeed, we found a strong correlation between those parameters (Fig 5E, 5F), suggesting that wound size dictates the number of cells that need to be replenished, and therefore the amount of proliferation and migration velocity of surrounding keratinocytes.”*

To answer the question in how far this correlation might be due to epidermal or dermal involvement, the removal of a substantial piece of epidermis without major interference of the dermal compartment would be required. This requires a systematic new experimental set-up and approach, which would completely change the focus of our study. Therefore, we feel that an even more detailed analysis is beyond the scope of our study.

3) The study uses a generic actin-CreERT2 mouse model to label keratinocytes, excluding the possibility to label basal or suprabasal cells selectively. The conclusions regarding specific basal/suprabasal responses would be strengthened if authors would look at each cell population independently.

Reply:

We agree with the reviewer that the use of a layer-specific Cre line, such as K14-CreERT2 or K10-CreERT2 for the basal layer or the suprabasal layer, respectively, would be the most elegant approach to categorically label keratinocytes. However, this requires an enormous time and money investment to import these mice in our animal facility and additionally breed them with mTmG mice. Therefore, for the revised manuscript we have chosen a less elegant, but still effective alternative approach to discriminate basal and suprabasal cells based on differential sizes of these cells and their position along the axial axis (see below for details).

First, we aimed to test whether basal and suprabasal cells have distinct cell sizes, and z-position along the axial axis. We determined cell sizes and Z-position in mice in which E-Cadherin coupled to the CFP fluorophore marks cells-cells contacts (E-cad-CFP mice) (**rebuttal letter Fig1**). We determined the K14-positive basal layer and K10-positive suprabasal layer by immunofluorescence, and used an unbiased machine learning approach to segment each cell (based on E-cad-CFP) and to measure cell areas (**rebuttal letter Fig1**). Importantly, this analysis revealed that basal and suprabasal cell populations have significant different cells sizes, and are located at different positions at the axial axis, and therefore can therefore be distinguished by size and z-position.

Based on this analysis, we observed that cell areas in both basal as well suprabasal layer stay near constant over time, with the same ratio of basal layer cells being smaller than suprabasal cells and that these differences are independent of the distance to the wound (**Fig 2A, 2B** and **Supplementary Fig S2B** of the revised manuscript).

Based on these criteria, we have reanalyzed the migration of basal and suprabasal cells in mTmG mice confirming our initial conclusions: basal cells migrate towards the wounds whilst suprabasal cells are rather static (**Fig 2F and H** of the revised manuscript).

To explain this approach to the reader, we have included this data and text at page 6 and 7 of the revised manuscript: “*To be able to faithfully determine and distinguish between basal and suprabasal layer upon wounding, we performed scratch wounds in E-cad-CFP mice. We identified the Keratin 14-positive basal and Keratin 10-positive suprabasal layer by immunofluorescence stainings on wounded skin whole mounts (supplementary Fig S2A). Using an unbiased machine learning approach, we used E-cad-CFP-signal to segment cells located at the basal and suprabasal layers. This analysis revealed that the more columnar and cuboidal basal keratinocytes are smaller ($50\text{-}300\ \mu\text{m}^2$) than the differentiated polygonal shape and flattened suprabasal keratinocytes ($\sim 300\text{-}600\ \mu\text{m}^2$) (Fig 2B). In both layers, the size of cells do not significantly change when located at different distances from the wound (Fig 2A and supplementary Fig S2B). Moreover, these differences in size between basal and suprabasal cells remain constant over time during wound healing (Fig 2B), illustrating that we can distinguish both layers based on the size of the cells.*”

Rebuttal Figure 1: Basal and suprabasal cells can be distinguished based on cell size and axial position.

A) Representative x-z-projections of immunofluorescence staining of skin whole mounts in E-cad-CFP mice immediately post wounding (upper panel). Representative images of E-cad-CFP signals in the basal or suprabasal layer, respectively (middle panel). This signal was submitted to unbiased segmentation and cell size measurement using ImageJ (lower panel). All scale bars $50\ \mu\text{m}$. **B)** Measurements of cell areas of basal (B, magenta) and suprabasal cells (SB, yellow), directly after wounding. **C)** Representative measurement of fluorescent intensity of immunofluorescent stainings on skin whole mounts (left panel) for K10 (suprabasal layer; yellow) and K14 (basal layer; magenta) along the Z-axis (right panel).

4) Previous in vitro scratch wound assays establishes proliferation as a major factor in wound healing. The authors do not comment on proliferation in their work, but it would be relevant to investigate if proliferation and migration correspond or influence each other. The material presented in the manuscript would allow for an investigation of proliferation by correlating clone size and migration patterns.

Reply:

In the revised version of the manuscript we now included the analysis of proliferation and the results on how proliferation and migration correspond to each other. To investigate proliferation, we performed live imaging of scratch wounds in Fucci2 mice (**Fig 5** of the revised manuscript). Cells in G1-phase are marked by the expression of nuclear mCherry-hCdt1, while proliferating cells in S/G2-phase are marked by the expression of nuclear mVenus-hGem. As shown in **Figure 5B-C** of the revised manuscript, we identify a zone of enriched proliferation 200-400 μm away from the wound side and this zone stays constant over the time of wound closure. Analysis of migration velocity revealed that proliferating cells (S/G2) migrate with comparable characteristics to non-proliferating cells (G1) (**Fig 5G** of the revised manuscript).

Our results suggest that the initial response to close the wound rapidly is executed by fast migrating basal keratinocytes directly adjacent to the wound, and that the replenishment of cells that are lost upon wounding is mediated by proliferation further away from the wound.

These new findings are described in the revised version of the manuscript and depicted in **Figure 5** at page 11: *“In order further characterize this proliferation in time and space in detail, we imaged repair of scratch wounds in Fucci2 mice (Abe et al, 2013). In these mice, individual cells can be identified and tracked based on their differential expression of mCherry-hCdt1 (magenta) in cells that are in a G1-cell cycle state and mVenus-hGem (green) in proliferating cells in S/G2 phase (Fig 5B, supplementary Movie 4). Using intravital microscopy starting 16 h post scratch wounding, we identified that ~ 5% of cells proliferate directly next to the wound site (0-200 μm), however the majority of proliferating cells (~ 10%) are localized in a zone 200-400 μm away from the wound side (Fig 5B, 5C). Within this zone the percentage of proliferating cells stays stable during the course of wound healing (Fig 5D).”*

And page 12: *“The observed persistence of a proliferative zone suggests that proliferation might be uncoupled from the response of keratinocytes to quickly migrate towards the wound bed with high velocity. To test this, we tracked individual cell in G1- or S/G2-phase, respectively, in the basal layer of the epidermis in different distances towards the wound bed. We found that proliferative (S/G2) and none-proliferative (G1) basal cells migrate with the same velocity (Fig 5G). Therefore, the proliferative state of a cell does not influence its ability to migrate.”*

5) The authors argue that re-epithelialization after scratch wound is independent of hair follicle stem cell contribution. However, supporting movie 3 shows what appears to be a labelled cell appearing from a hair follicle integrating into the basal layer. More data would be needed to support the claim that there is no HFSC contribution to epidermal wound healing in this model.

Reply:

In order to strengthen our point, in the revised manuscript we now included a quantification of migration over the course of scratch wound healing of labeled cells within the upper part of the hair follicle (upper isthmus and infundibulum) in comparison to surrounding basal cells in different distances away from the wound site (**Figure 4B, 4C** of the revised manuscript). As depicted in **Figure 4B and 4C**, in contrast to surrounding basal cells that move towards the wound, cells within the hair follicle do not move out of the hair follicle but rather stay associated with the hair follicle independent of their proximity to the wound.

We included this important data in **Fig 4B, 4C** and describe this data at page 11 of the revised manuscript: *“Using our mTmG model, we imaged hair follicles in different distances to the wound. We never observed GFP⁺ cells migrating out of a hair follicle. Independent of the hair follicle distance to the wound, GFP⁺ cells within the infundibulum and isthmus upper part of the hair follicle did not migrate at all and stayed attached in the hair follicle, while hair follicle-surrounding basal cells followed their migration track towards the wound, suggesting a hair follicle-independent mode of scratch wound healing (Fig 4B, 4C).”*

6) The authors claim that there are no alterations in cell size observed during scratch wound healing. No data is presented to support this claim, which would be an important point to establish since all conclusions are based on cell size and localization instead of established basal/suprabasal markers.

Reply:

We agree with this comment. As explained in our reply to **comment 3** of this reviewer (see above), in the revised manuscript we now present data that shows that sizes of cells in both basal as well suprabasal layer stay near constant over time, with the same ratio of basal layer cells being smaller than suprabasal cells and that these differences are independent of the distance to the wound (**Fig 2A-B** and **Supplementary Fig S2B** of the revised manuscript).

Minor points:

7) Figure 1 -

The authors argue that there is no dermal injury in this model, however in figure 1C it looks like there is dermal reorganization at 8 and 16 h after injury. Could dermal markers be used to confirm or dismiss?

It looks like Krt6 is uniformly high in the epidermis directly after injury. Krt6 is normally expressed upon stress or injury, but undetectable in normal epidermis. Please clarify.

Reply:

To address this point, we included stainings for fibronectin as a read-out for dermal remodeling as shown in **Figure 1C** and state our findings at page 5 of the revised manuscript.

We very much appreciate the notion that our Keratin 6 staining is not convincing. We therefore repeated the staining using a fresh commercial antibody (see Materials and Methods) and also included a Keratin 6 staining of unwounded skin, indeed showing that Keratin 6 is locally upregulated upon wounding but absent in the unwounded skin (see new **Figure 1D**).

8) Figure 2-

In figure 2C it would be nice to show how close to the wound bed the images are taken,

especially since authors argue that the migration pattern is depending on the distance to the wound. It is also not clear why these specific time points are chosen.

Reply:

In the revised version of the manuscript we included a sentence explaining the chosen time point for imaging at page 8: “Starting at 16 h post wounding and imaging for 8 h would cover both times before and after, respectively, the first basal keratinocytes would enter the wound bed.”. We also included the information in the figure legend that new **figure 2E** represents migration pattern of keratinocytes 290 μm away from the wound.

9) In figure 2D authors display cell size and migratory velocity. Based on the previous publications and wound models (Park et al 2017), it is not impossible that basal and/or suprabasal cells alter the cell size when becoming migratory. This would scew the analysis - perhaps authors could compare the size of migratory and non-migratory cells to rule this out.

Reply:

Please see our reply to **comment 3** how we have addressed this point.

10) It is not entirely clear what figure 2F represents - how is directionality measured, and what would be the rationale for suprabasal cells to have such a strong trend against migrating towards the wound site? How many cells were assessed for directionality of each mouse?

Reply:

We apologize for our unclear phrasing of our point. We intended to explain that although some cell movement is observed in the suprabasal layer, this movement is random and not directed toward the wound, so that cells remain close to their initial position.

We do now realize that the directionality measurement may be a complicated manner to illustrate our point. For the revised manuscript, we replaced these measurement for rose plots in which we depicted the movement pattern of representative individual cells located in the basal and suprabasal layer. This rose plots are shown at **Figure 2H** and described at page 8/9 of the revised manuscript: “To test whether the directionality of migration towards the wound is different for basal and suprabasal cells, we constructed rose plots of representative positions (Fig 2H). While the majority of suprabasal keratinocytes migrated little and displayed a non-directional random movement, basal keratinocytes showed prominent directed movement towards the wound site.”

11) The leap-frog model suggests that suprabasal cells can move down and fill up the basal layer, and authors argue that this does not happen in this injury model. How can authors exclude this possibility with no layer specific labelling? Size is used as the only determinant of cell state, and it is possible that cell size will depend on location within the epidermis.

Reply:

As explained in detail in our reply to comment 3, our newly performed measurement on cell size illustrate that cell size is not affected upon wounding, independent of the cell position towards the wound. Second, as explained in our reply to comment 10, the slow movement of suprabasal cells is random and not towards the wound (see **Figure 2H** of the revised manuscript). Third, during our analysis we put extra attention on the reviewers note, but we could never observe suprabasal cell migration into the wound bed.

12) Figure 3-

Here the authors follow labelled pairs to determine migration trajectories of cell originating in close proximity. The existence of pairs indicates that labelled cells have proliferated after tamoxifen administration (see previous comments on proliferation). Figure 3C indicates that labelled pairs closer to the wound are more likely to separate from each other over time - is this significant compare to the other cell populations measured? And does increase reorganization correlate to proliferation patterns?

Reply:

To address this point, we performed statistical tests on our data. This showed that, indeed, the separation of labelled pairs closer to the wound is significant from 2h onwards. In the revised manuscript, we have added statistical analysis in **Figure 3 and supplementary Figure 3**.

Pairs of labelled cells might have indeed originated from proliferation. However, also the random labelling approach can give rise to labelling of cells that are direct neighbors. Based on our additional data on proliferation (as described above), we show that proliferating cells (S/G2) migrate with a similar velocity and displacement compared to non-proliferating (G1) cells and that proliferation takes place in proliferative zone 200-400 μm adjacent to the wound. However, the migration velocity in this zone is comparably slower than directly adjacent to the wound site. We suggest, that proliferation might induce a local tissue crowding that might lead to an overall decrease of migration velocity within the proliferation zone.

13) In Supplemental figure 2 authors show that the speed of migration is dependent on distance to wound bed - is this change statistically significant? If migration speed is higher closer to the wound, labelled cells would be expected to be further away from each other compared to cells further away from the wound.

Reply:

The reviewer may be correct that one of the mechanism of labelled cells to depart may be the increased migration velocity. This may therefore contribute to the swarm behavior as described in **Fig 4A**. And yes, the change is statistically significant. In the revised manuscript we have added the statistical test to the figure (see **Supplementary Fig 3**).

14) Movie 1: seems to run for up to 40h instead of the 20h stated in the figure legend.

Reply:

Thanks for pointing out this typo, which we have corrected in the revised manuscript.

Reviewer #3 (Comments to the Authors (Required)):

The authors develop an in vivo partial thickness skin wounding model in mice, in an attempt to provide the first real-time data as to the cellular mechanism of closure. The manuscript does a good job motivating why this is interesting, contrasting with the known biology of full thickness wounds and presenting the imaging data. I think there is a significant contribution here in terms of demonstrating the relatively higher study of basal keratinocytes to motility and the high degree of fluidity and neighbor exchanges among the basal keratinocytes, especially those moving the fastest, closest to the wound edge. I think this study will provide a useful method and new conceptual framework for the field.

Reply:

We thank this reviewer for his/her time to review our manuscript, and are happy that this reviewer values our work as a significant contribution to the field.

1) I have one substantive criticism: the authors state that this fluidity enables them to bypass immobile structures (e.g. hair follicles) and I agree that it could. But I didn't see where in their figures they observed such a fluid migration around an immobile structure. Accordingly, it should not be presented as a conclusion but rather a speculation or a possibility for future investigation.

Reply:

We appreciate that our data in our initial version of the manuscript did not convincingly illustrate the fluid migration around immobile structures.

In order to test this better, we imaged the location and movement of each nuclear labelled cell upon scratch wounds in the Fucci2 mouse model. These imaging experiments indeed illustrated that basal cells bypass hair follicle while migrating towards the wound (see **supplementary movie 3** and **Figure 4A** of the revised manuscript). In the stills of this movie (**Figure 4A**) we annotated the migration trajectory of each cell, which illustrates that cells migrate around immobile hair follicles.

We show this data in **Fig 4A** and **supplementary movie 3**, and describe this data at page 10 of the revised manuscript: "*Keratinocytes moving towards the wound may potentially be blocked by hair follicles. Therefore, we wondered whether the swarm migration behavior of interfollicular basal keratinocytes enabled them to bypass hair follicles. We used intravital microscopy on scratch wounds in fluorescent ubiquitylation-based cell cycle indicator 2 (Fucci2) mice in which the nucleus of cells is fluorescently labelled. We found that keratinocytes bypassed hair follicles without altering overall directionality (Fig 4A and supplementary Movie 3). Basal keratinocytes on their way towards the wound side that arrived at the point of a hair follicle exchanged their neighbors to circumvent the obstacle and to continue migrating in their direction (Fig 4A).*"

Minor:

1. Abstract: "behave rather passively" should be either "are rather passive" or "behave passively".

Reply:

Thanks for pointing out this type, that we have corrected in the revised manuscript.

2. Page 9: "behave uniform" should be "behave uniformly".

Reply:

In the revised manuscript we have also corrected this typo.

3. There is something wrong with the axis labels in many of the graphs. When printed, many letters were not visible and the labels could not be interpreted.

Reply:

We apologize for this, which most likely happened during the conversion to pdf. For the revised manuscript, we have double checked all axis labels, and hope that all the labels are shown correctly in the pdf of the revised manuscript.

November 15, 2020

RE: Life Science Alliance Manuscript #LSA-2020-00765-TR

Prof. Jacco van Rheenen
Onco Institute, The Netherlands Cancer Institute
Molecular Pathology
Plesmanlaan 121
Amsterdam, I am not in the U.S. or Canada 1066 CX
Netherlands

Dear Dr. van Rheenen,

Thank you for submitting your revised manuscript entitled "Scratch-induced partial wounds re-epithelialize by sheets of independently migrating keratinocytes". We would be happy to publish your paper in Life Science Alliance pending final revisions in accordance to minor comments from the reviewers (see below) and necessary to meet our formatting guidelines.

Along with the points listed below, please also attend to the following,

- please add a callout for Fig 1B
- please upload Fig 6 as a separate file, similar to others
- please update Dr. Morgner's account with their ORCID ID, Dr. Morgner should have already received instructions in their email for how to do it
- please add the legend for Fig 4C

A. FINAL FILES:

- An editable version of the final text (.DOC or .DOCX) is needed for copyediting (no PDFs).
- High-resolution figure, supplementary figure and video files uploaded as individual files: See our detailed guidelines for preparing your production-ready images, <https://www.life-science-alliance.org/authors>
- Summary blurb (enter in submission system): A short text summarizing in a single sentence the

study (max. 200 characters including spaces). This text is used in conjunction with the titles of papers, hence should be informative and complementary to the title. It should describe the context and significance of the findings for a general readership; it should be written in the present tense and refer to the work in the third person. Author names should not be mentioned.

B. MANUSCRIPT ORGANIZATION AND FORMATTING:

Sincerely,

Shachi Bhatt, Ph.D.
Executive Editor
Life Science Alliance
<https://www.lsjournal.org/>
Tweet @SciBhatt @LSAJournal

Reviewer #1 (Comments to the Authors (Required)):

this manuscript now answers all my comments and I am happy for publication

Reviewer #2 (Comments to the Authors (Required)):

The revised version of the manuscript is significantly improved and would be suitable for publication. I have some minor comments that would potentially be nice to clarify before being accepted.

The authors state in the text

"We identified by 5-ethynyl-2'-deoxyuridine (EdU) incorporation that most proliferation was present 16 h post scratch wound initiation (Fig 5A)."

How is EdU incorporation quantified? Is HF proliferation also increased, and if so, what happens to these cells if they do not contribute to wound healing?

The authors state in the text

"However, the loss of cells upon wounding needs to be fueled by proliferation."

Perhaps the intention of the sentence is rather to state that cells need to be replenished by proliferation?

The authors state in the last sentence that re-epithelialization of superficial wounds provides a degree of plasticity to efficiently and rapidly cover differently shaped wounds. It is my understanding from the manuscript that it is the size, rather than the shape, that is being analyzed.

Is the velocity of S vs. G1 cells also directed movement? Perhaps it is possible to clarify?

Figure 6 is missing in the manuscript, although referred to in the text.

Movie1 - layers titled "suprabasal" and "basal", should be "suprabasal" and "basal"

Reviewer #3 (Comments to the Authors (Required)):

The authors were highly responsive in revision. I very much appreciate the revised Figure 4A. I am fully supportive of publication. I note one minor issue in phrasing that might be coming across differently than intended. As constructed, I read the sentences below that proliferation is fueling (read causing) cell loss (death). This seems unlikely to either be true or what the author meant. I suggest a different phrasing below.

"The loss of cells due to damage is fueled by proliferation in a distinct zone away from the wound site and proliferation does not affect overall migration pattern" (p4). As written, this sentence implies that proliferation is killing cells. I think you mean something like "The cells lost due to damage are replaced by proliferation..." Likewise with "However, the loss of cells upon wounding needs to be fueled by proliferation." (p16).

Reviewer #1 (Comments to the Authors (Required)):

This manuscript now answers all my comments and I am happy for publication

Reply:

We would like to thank this reviewer for his/her support to publish our manuscript.

Reviewer #2 (Comments to the Authors (Required)):

The revised version of the manuscript is significantly improved and would be suitable for publication. I have some minor comments that would potentially be nice to clarify before being accepted.

Reply:

We also thank this reviewer for his/her support to publish our manuscript. Below we explain how we have addressed the last few comments.

The authors state in the text

"We identified by 5-ethynyl-2'-deoxyuridine (EdU) incorporation that most proliferation was present 16 h post scratch wound initiation (Fig 5A)."

How is EdU incorporation quantified? Is HF proliferation also increased, and if so, what happens to these cells if they do not contribute to wound healing?

Reply:

We thank this reviewer for pointing this out. We quantified EdU incorporation by counting the EdU positive cells of the interfollicular epidermis basal cells per timepoint. We added this quantification in the manuscript in Figure 5A (see right panel).

The authors state in the text

"However, the loss of cells upon wounding needs to be fueled by proliferation."

Perhaps the intention of the sentence is rather to state that cells need to be replenished by proliferation?

Reply:

We apologize for our unclear wordings, and indeed we meant to state that the loss of cells need to be replenished by proliferation. At page 11 of the revised manuscript, we adapted the sentence accordingly which now reads: "*However, the loss of cells upon wounding needs to be replenished by proliferation.*".

The authors state in the last sentence that re-epithelialization of superficial wounds provides a degree of plasticity to efficiently and rapidly cover differently shaped wounds. It is my understanding from the manuscript that it is the size, rather than the shape, that is being analyzed.

Reply:

We agree and adapted the sentence accordingly to : "We show for the first time that a patterned collective behavior occurs *in vivo* during re-epithelialization of superficial wounds providing the degree of plasticity not only to efficiently and rapidly cover differently shaped superficial wounds of different diameters but also to ensure that cells reach the wound even in the presence of obstacles." at page 13 of the revised manuscript.

Is the velocity of S vs. G1 cells also directed movement? Perhaps it is possible to clarify?

Reply:

Cells migrate with a high directionality towards the wound regardless of whether cells are in the S or G1 state (see suppl Movie 4). We did not see a significant difference in migration persistence (a measure for directed movement) between cells that are in the different states. We have included this analysis in Fig 5H and describe this at page 12 of the revised manuscript: *"We found that proliferative (S/G2) and non-proliferative (G1) basal cells migrate with the same velocity (Fig 5G) and directionality (Fig 5H). Therefore, the proliferative state of a cell does not influence its ability to migrate."*

Figure 6 is missing in the manuscript, although referred to in the text.

Movie1 - layers titled "suprabasel" and "basel", should be "suprabasal" and "basal"

Reply:

Thanks for pointing this out. We corrected it in the revised version of the manuscript.

Reviewer #3 (Comments to the Authors (Required)):

The authors were highly responsive in revision. I very much appreciate the revised Figure 4A. I am fully supportive of publication. I note one minor issue in phrasing that might be coming across differently than intended. As constructed, I read the sentences below that proliferation is fueling (read causing) cell loss (death). This seems unlikely to either be true or what the author meant. I suggest a different phrasing below.

Reply:

We also thank this reviewer for his/her support to publish our manuscript in Life Science Alliance.

"The loss of cells due to damage is fueled by proliferation in a distinct zone away from the wound site and proliferation does not affect overall migration pattern" (p4). As written, this sentence implies that proliferation is killing cells. I think you mean something like "The cells lost due to damage are replaced by proliferation..." Likewise with "However, the loss of cells upon wounding needs to be fueled by proliferation." (p16).

Reply:

We apologize for our bad wording. In the revised manuscript, we have corrected this sentence accordingly and it now reads: "The loss of cells due to damage is replenished by proliferation in a distinct zone away from the wound site and proliferation does not affect overall migration pattern." and the sentence on page 11 to "Immediate keratinocyte migration towards the wound is crucial to close the wound side. However, the loss of cells upon wounding needs to be replenished by proliferation."

November 18, 2020

RE: Life Science Alliance Manuscript #LSA-2020-00765-TRR

Prof. Jacco van Rheenen
Onco Institute, The Netherlands Cancer Institute
Molecular Pathology
Plesmanlaan 121
Amsterdam, I am not in the U.S. or Canada 1066 CX
Netherlands

Dear Dr. van Rheenen,

Thank you for submitting your Research Article entitled "Scratch-induced partial wounds re-epithelialize by sheets of independently migrating keratinocytes". It is a pleasure to let you know that your manuscript is now accepted for publication in Life Science Alliance. Congratulations on this interesting work.

DISTRIBUTION OF MATERIALS:

Again, congratulations on a very nice paper. I hope you found the review process to be constructive and are pleased with how the manuscript was handled editorially. We look forward to future exciting submissions from your lab.

Sincerely,

Shachi Bhatt, Ph.D.

Executive Editor

Life Science Alliance

<https://www.lsjournal.org/>
